# Impact of Culinary Medicine Course on Confidence and Competence in Diet and Lifestyle Counseling, Interprofessional Communication, and Health Behaviors and Advocacy

**DOI:** 10.3390/nu15194157

**Published:** 2023-09-26

**Authors:** Britta Retzlaff Brennan, Katherine A. Beals, Ryan D. Burns, Candace J. Chow, Amy B. Locke, Margaret P. Petzold, Theresa E. Dvorak

**Affiliations:** 1Department of Nutrition and Integrative Physiology, University of Utah, Salt Lake City, UT 84112, USA; katherine.beals@hsc.utah.edu; 2Department of Health and Kinesiology, University of Utah, Salt Lake City, UT 84112, USA; ryan.d.burns@utah.edu; 3Department of Internal Medicine, University of Utah, Salt Lake City, UT 84112, USA; candace.chow@hsc.utah.edu; 4Department of Family and Preventative Medicine, University of Utah, Salt Lake City, UT 84112, USA; amy.locke@hsc.utah.edu (A.B.L.); tricia.petzold@hsc.utah.edu (M.P.P.)

**Keywords:** culinary medicine, nutrition education, interprofessional education, teaching kitchen, team-based learning, healthcare professional students, lifestyle medicine counseling, medical training, nutrition counseling, nutrition care

## Abstract

Most physicians report inadequate training to provide diet and lifestyle counseling to patients despite its importance to chronic disease prevention and management. To fill the nutrition training gap, elective Culinary Medicine (CM) courses have emerged as an alternative to curriculum reform. We evaluated the impact of an interprofessional CM course for medical and health professional students who experienced the hands-on cooking component in person or a in mixed-mode format (in-person and via Zoom) at the University of Utah from 2019–2023 (*n* = 84). A factorial ANOVA assessed differences between educational environment and changes between pre- and post-course survey responses related to diet and lifestyle counseling, interprofessional communication, and health behaviors and advocacy. Qualitative comments from post-course surveys were analyzed on a thematic level. Students rated themselves as having greater confidence and competence in diet and lifestyle counseling (*p* < 0.05) and increased ability to prepare eight healthy meals (*p* < 0.05). Additionally, a Mann–Whitney two-sample rank-sum test was used to compare data from exit survey responses from medical students who took the CM course (*n* = 48) and did not take the CM course (*n* = 297). Medical students who took CM were significantly more likely to agree that they could counsel patients about nutrition (*p* < 0.05) and physical activity (*p* < 0.05). CM courses may improve students’ confidence to provide diet and lifestyle counseling.

## 1. Introduction

In the United States, poor quality diet is the most significant risk factor for the most common chronic diseases (e.g., heart disease, type 2 diabetes, stroke, and certain types of cancer) [1]. Furthermore, their interrelated etiology supports comorbidity; the CDC estimates that 60% of US adults have one chronic disease, whereas 40% have two or more [1,2]. These conditions are the leading causes of premature death and disability and the leading drivers of the United States’ 4 trillion dollars in annual healthcare costs [3]. The socio-economic burden on the nation’s financial and healthcare systems cannot be overstated. 

Unfortunately, diet and lifestyle counseling does not occur during clinical care in proportion to the prevalence of these chronic, diet-related conditions [4,5,6]. Recent estimations show that dietary counseling by physicians occurs in only 20–40% of patient visits [4,5,6]. Consistent failure to address diet in a clinical context has been described as an ethical lapse [7]. Considering the potency of dietary influences on health, many in the medical and public health fields agree that doctors should be trained and supported to assist patients with initiating positive lifestyle changes—specifically those involving diet and exercise.

Physicians cite several factors that prevent them from engaging with patients about diet and health, including lack of knowledge, training, time, and self-efficacy to engage in diet-related counseling [8,9,10]. This lack of preparedness stems from the well-documented deficit of nutrition education during medical training [11,12,13,14]. Several reasons for the low prioritization of nutrition in medical training have been recognized. They include a greater focus on technologically advanced acute and chronic treatments, a shortage of core nutrition faculty, and an already crowded curriculum [15]. Each reason may be considered separately, but they all reflect the greater institutional inertia that has yet to be overcome. 

Many organizations and agencies have called for nutrition competencies to be prioritized in the curriculum for medical students and other healthcare professionals, including the American Heart Association; the Bipartisan Policy Center; the National Heart, Lung, and Blood Institute; the Academy of Nutrition and Dietetics; and the White House [16,17,18,19]. They all point to the massive body of evidence accumulated over the last several decades, including diet interventions for treating chronic diseases such as diabetes and hypertension; interventions that compete with pharmacologic interventions in their effectiveness, often with less risk, reduced side effects, and lower costs [20,21]. 

Though all physicians should be prepared to discuss diet-related questions with patients, the primary care experience has been identified as a prime opportunity to help people achieve a healthier diet [22]. Primary care physicians (PCPs) are ideally placed as an initial point of contact within the healthcare system, and a high proportion of adults and children have contact with a PCP each year. Not only do PCPs have convenient nutrition assessment and education opportunities, they are uniquely positioned to refer and collaborate with Registered Dietitian Nutritionists (RDNs) and reinforce nutrition care provided by RDNs. Moreover, physicians are perceived as highly credible sources of health information [23,24]. Of those who received diet information from their doctors, 78% reported changing their eating habits due to those conversations [25]. Analysis of NHANES data found overweight and obese patients whose doctors spoke to them about weight were twice more likely to lose >5% body mass over the following year [26]. Similarly, Rose et al. concluded that PCP advice on weight loss significantly impacted patient attempts to change behaviors related to their weight [27]. 

Medically trained clinicians are well-placed to initiate nutrition care and must be supported to acquire necessary nutrition competencies. To that end, Culinary Medicine (CM) programs have emerged as an innovative education movement in lieu of top-down medical school curriculum reform. CM courses are designed to equip participants with the knowledge and skills to translate the science of nutrition into healthy lifestyle practices that prevent disease and improve health. They customarily take place in a teaching kitchen (TK), a physical or virtual venue that serves as an educational classroom and learning laboratory for food-based experiential learning [28]. Similar to other topics in medical education, CM training embraces the model of simulation-based medical education with deliberate practice (SBME-DP) and an active learning culture (AL). Rooted in Ericsson’s conceptual framework of deliberate practice, SBME-DP is part of a wider trajectory of curricular and pedagogical reform in medical education [29]. Studies suggest that SBME-DP is superior to traditional clinical medical education in acquiring clinical skills [30,31]. Likewise, recent reform efforts in healthcare education have also emphasized the value of AL to shift away from passive knowledge acquisition toward improved student engagement and critical thinking [32]. AL is an umbrella term for various teaching and learning techniques such as case-based learning, experiential learning, peer problem-solving, problem-based learning (PBL), etc. The specific AL components of CM courses often include a flipped classroom, hands-on cooking labs, counseling simulations, group discussions, and case-based learning. 

CM programs appear to be a feasible replacement for traditional didactic nutrition education and may be more effective at improving student nutrition competencies [33]. Medical students who completed CM courses reported increased confidence in counseling patients on a healthy diet and other positive outcomes, such as increased familiarity with evidence-based nutrition interventions [34,35,36,37]. They also reported a better understanding of the role of dietitians in patient care and thus may have become greater proponents of nutritional care and sources of referral [31]. Compared to those receiving traditional nutrition education, CM trainees expressed higher competency in counseling for lifestyle medicine topics [38]. At Tulane University, a large prospective, observational cohort study demonstrated that CM education improved students’ dietary patterns and attitudes about the efficacy of nutrition counseling compared to traditional nutrition education [31]. CM courses also help students cook and eat healthily [38]. Importantly, as medical and health professional students are empowered to cook and eat healthfully, they will be more likely to counsel their patients to do the same. There exists a solid evidence base for this translational effect. Researchers continually find that primary care physicians’ strongest predictors of health promotion counseling are practicing healthful behavior themselves [39,40,41,42,43,44].

These positive findings help to explain why the implementation of this pragmatic participatory approach is gathering momentum around the country in medical schools and as a continuing medical education (CMEE) opportunity [45]. As of 2022, 60 academic medical centers have adopted the *Health Meets Food* curriculum developed by the Goldring Center for Culinary Medicine at Tulane to offer CM training [46]. 

Beyond medical school, programs targeting continued medical education are introducing CM educational opportunities, attuning to the opportunity to deliver a relatively low-cost, potentially high-impact intervention to professionals who have the capacity to influence patient behavior changes. Currently, CM courses are diverse in format, duration, instructors, and dietary approach [33,45]. Though this diversity presents challenges to researchers when comparing the impact of CM interventions, this adaptability means that nearly any educational setting can provide a CM course if institutional support exists. 

The primary purpose of this study was to evaluate the impact of the University of Utah’s Culinary Medicine course on improving perceived confidence and competence in dietary counseling, lifestyle medicine counseling, interprofessional communication, and health behaviors and advocacy. A secondary purpose was to compare pre-course and post-course changes for the different cooking lab groups (in-person vs. online/hybrid.) A tertiary purpose was to evaluate the Transition to Internship (TTI) survey responses from fourth year medical students and compare the students who completed the Culinary Medicine course to those who did not participate. 

## 2. Methods

### 2.1. Subjects for Pre/Post-Course Survey and TTI Survey

The subjects for the pre-/post-course survey included 84 University of Utah students who completed the U of U Culinary Medicine (CM) course and completed a pre-and post-course survey during the Fall or Spring semesters of 2019–2023. (Table 1). Due to its elective nature, students self-selected into the course.

Subjects for the TTI Survey included 345 fourth-year medical students from the University of Utah School of Medicine in academic years 2019–2020 and 2021–2022.

### 2.2. CM Course Curriculum and Structure

At the University of Utah (U of U), the Department of Nutrition and Integrative Physiology and the Department of Family and Preventative Medicine launched an 8-week CM course in 2016, intending to strengthen future clinical care with nutritional and team-based approaches. Like CM courses offered at other universities, the U of U CM course takes place in a teaching kitchen, where students actively prepare recipes together. Initially, instructors, which included a dietitian, a physician, a chef, and a graduate assistant, launched the course utilizing the *Health Meets Food* curriculum. Beginning in the Fall 2017, however, the CM instructors developed a novel course curriculum to make adaptations they considered relevant to the course objectives. (Table 2). At this time, it also became an interprofessional education (IPE) course offered to medical students and other health science graduate students. Other adaptations included an expanded recipe collection, incorporating medically diverse case studies, role-playing, motivational interviewing for students to practice counseling skills in a safe, non-judgmental environment, and assessments/quizzes tailored to all health professions students. Additionally, the curriculum underwent a major revision in 2020 to provide learners with the most up-to-date course material, information, and literature. Instructors also made organizational changes based on student feedback by including more culturally diverse recipes, moving the case study discussion online, and increasing the presence of other programs of study, specifically nursing students. 

The U of U CM course is a recurring 8-week IPE course available to medical and other health professional students in the Fall and Spring semesters. Each iterative weekly session has a topic integrated across the didactic, cooking, and discussion portions. Medical doctors and registered dietitians with extensive cooking experience team-teach the course. The structure of the weekly coursework involves three main components: (1) online didactic pre-work, (2) hands-on cooking lab with interactive discussions on nutrition topics, and (3) patient case study discussions. Before the weekly lab, students completed approximately one hour of online learning (voice-over PowerPoint lectures, required reading, quizzes, food demonstration videos, and other required items). During the 2 h weekly lab, students participated in 90 min of supervised cooking in small groups, followed by 30 min of a shared meal and an interactive discussion. The weekly case study discussions took place in the classroom or online. See Table 2 for the entire course outline. 

For example, in Week 2, students learned about behavior changes and assessing patient diet history. They watched a 12 min presentation on the basics of facilitating behavior changes and a 20 min motivational interviewing (MI) video, in which a dietitian interviewed a patient utilizing the MI technique. The required reading for the week included a journal article titled “Collaboration and Negotiation: The Key to Therapeutic Lifestyle Change,” which elucidated the effectiveness of the coach approach for chronic disease management rather than the expert approach often used in acute care settings. Additionally, students were required to watch four 1–3 min videos demonstrating culinary techniques, including preparing soft-boiled eggs, pickling onions, slicing avocado, and stemming and chopping jalapenos. For the hands-on lab, students prepared various recipes such as Tomatillo Salsa, Black Bean Tostadas with Quick Pickled Veggies, Soft-cooked Eggs, and Chicken Posole. The case study component consisted of students’ reviewing a particular patient and diet history, to which they responded to the following prompts: consider what condition(s) the patient had that would improve with a more minimally processed plant-forward eating pattern; practice counseling the patient using the 5 A’s: Ask, Assess, Advise, Agree, and Arrange; and help the patient set SMART goals (Specific, Measurable, Achievable, Relevant, and Timebound).

### 2.3. Course Theoretical Framework

Utilizing the SBME-DP approach, the CM course engages learners in structured and repeated learning opportunities that are experiential in nature. With peer support and instructor guidance, students engage in lifelike experiences to improve nutrition-related skills (e.g., simulated patient counseling related to nutrition and health behavior changes), culinary skills, recipe literacy, and the ability to prepare healthful meals. Additionally, learners are provided feedback during and following meal preparation, and instructors provide opportunities for self-reflection to help improve counseling confidence. 

The hands-on cooking and guided practice aim to foster the student’s self-efficacy regarding culinary skills, nutrition and lifestyle counseling, and their own personal dietary choices. Building on Bandura’s Theory of Self-Efficacy, individuals with high self-efficacy expectancies—or the belief that they can accomplish what they envision—become self-fulfilling [47]. They are more effective at achieving their goals than individuals with low self-efficacy expectancies. 

### 2.4. Pre- and Post-Course Surveys and Data Collection

The pre-course and post-course surveys were developed by a multidisciplinary team of dietitians and physicians and reviewed by a survey expert in health research. The surveys contained five categories of questions: (1) general course feedback, (2) dietary assessment and advice, (3) lifestyle counseling topics, (4) interdisciplinary communication, and (5) students’ health behaviors and wellness advocacy. Students scored their responses on a traditional 5-point Likert scale ranging from not at all competent/confident to highly competent/confident or strongly disagree to strongly agree. Personal health questions (e.g., cups of vegetables per day and hours of sleep per night) required students to select quantities and durations. The open-ended questions allowed students to type out responses with no character limit. The pre-course survey had twenty-three questions and was administered before coursework began in Week 1. Students received 10 points toward a pass/fail grade if they completed the pre-course survey. The post-course survey had thirty questions and was administered after the final coursework in Week 8. Like the pre-course survey, students received 10 points toward a pass/fail grade as an incentive for completion. Both surveys were administered online in the Canvas educational platform and took an estimated 8–10 min to complete. All survey responses were anonymous and compiled within Canvas. 

### 2.5. Transition to Internship Survey Data Collection

Additional data for this study were collected from the Transition to Internship (TTI) survey administered to 4th-year medical students in academic years 2019–2020 and 2021–2022. TTI is the final course in medical school training at the U of U. The survey questions pertained to the TTI course and the overall education experience at the U of U School of Medicine. The Office of Education Quality Improvement utilized the data for faculty review and advancement, School of Medicine accreditation, and to prioritize improvement areas. Surveys were sent via Qualtrics on the final day of class. Completion was required but could not be enforced since the students left the university. As such, the response rate was less than 100%. Students were eligible to take the CM course in any year of their medical training, with the majority taking it in their first, second, or fourth year. 

### 2.6. Independent Variables

Time. Survey questionnaires were completed at two timepoints: pre-course (timepoint 1) and post-course (timepoint 2), with completion of CM course occurring between surveys.

Educational Environment: In-person cooking lab vs. online/hybrid cooking lab. Due to the social distancing required during the COVID-19 pandemic, the U of U CM course integrated an Interactive Video Conferencing (IVC) synchronous learning environment while utilizing the student’s own kitchens. The U of U has a large teaching kitchen facility. However, meeting fully in person on campus or in the community was not possible starting the Fall 2020 semester. Therefore, instructors reworked the cooking portion of the course to be administered via the Zoom platform. For the first hybrid semester in Fall 2020, only six students were allowed to be in the teaching kitchen for each cooking session. The remaining students were required to participate from home. For the second hybrid semester in Fall 2021, students could cook in person on campus or via Zoom in their kitchens. For the Spring 2021 semester, students participated in cooking sessions exclusively via Zoom. All students who cooked at home performed their own shopping and preparation of ingredients. During class time, in Zoom breakout rooms, they individually prepared and discussed the recipes from their own kitchen spaces. To evaluate educational environment, the hybrid and online cooking groups were combined into a “mixed-mode” group (*n* = 37) to compare to the in-person cooking group (*n* = 47). Table 3.

### 2.7. Outcome Variables

Confidence and Competence in Dietary and Lifestyle Medicine Counseling. Dietary counseling is a skill set that combines dexterities in assessment, communication style, implementation, background knowledge, etc. Questions on the pre/post-course survey which asked about one of these aspects were included in a composite variable for dietary counseling. Similarly, any survey question that addressed a lifestyle medicine topic was included in a composite variable for lifestyle medicine counseling. Table 4 highlights the questions that were included in each composite variable.

Interprofessional Communication. The IPE nature of the CM course makes it unique among CM courses. The U of U CM course is open to all graduate health science students such as medicine, pharmacy, nursing, dentistry, occupational health, physical therapy, nutrition, and health coaching. Differences were compared for the survey question, “Students will be able to effectively communicate with other professionals in an interdisciplinary manner.”

Health Behaviors and Advocacy. Individual survey questions that asked students about their health habits such as vegetable consumption, meals eaten out, hours slept, physical activity, and ability to prepare 8 healthy meals were compared from pre/post-course. Additionally, survey questions that asked students about their areas of future growth in nutrition and food preparation or championing a healthy lifestyle for themselves or others comprised the remaining outcome variables. 

### 2.8. Design and Analysis

Data from the pre-post course surveys were analyzed using quantitative and qualitative methodology. Likert responses to the pre-and post-course survey questions were coded 1 through 5, from least favorable to most favorable, and were treated as continuous variables [48]. A 2 × 2 Factorial Analysis of Variance (ANOVA) test was used for the primary quantitative analyses. The effects of interest from the ANOVA model included the time main effect or changes pre- to post-course and the group × time interaction or differences between cooking groups. Bonferroni adjustment was made for multiple tests where appropriate, with adjustment to the initial alpha level set at *p* < 0.05. Statistical analyses for pre-post survey data were performed using Stata 17.0 Basic Edition statistical software package (StataCorp., College Station, TX, USA). 

For the qualitative analysis, we examined the open-ended survey questions in which students commented on what they liked best and what they liked least about the course. A thematic analysis was performed using the three elements of the differentiated instruction model as categories (content, process, and product) [49,50]. Any comments related to course materials, knowledge, or structure were considered content comments. Responses describing relational, experiential, or social aspects were considered process comments. Thirdly, when students described a change, outcome, or application, it was categorized as a product comment. Following this classification process, we conducted a frequency analysis of the key themes within each category.

Responses to the TTI survey data were analyzed by whether students took the CM course (*n* = 48) or did not take the CM course (*n* = 297). Responses were also analyzed by what year they took the course (during the first two pre-clinical years or the third and fourth clinical years). Mann–Whitney two-sample rank-sum tests were used to compare the Likert-scale items (e.g., I agree that I can counsel patients about nutrition, 5 = strongly agree, and 1 = strongly disagree), which were treated as ordinal data, between these independent samples (by whether students took the course and by year taken). Effect size r was calculated as *z* statistic divided by the square root of the sample size (N) (Z/√N).

## 3. Results

Eighty-four CM students completed both the pre-course and post-course surveys. Although no identifying student information was collected at the time of the survey, based on the registration ID, the overall student composition was determined to be 41 medical students FPP MD 7540-01) and 43 other health professional students (NUIP 7540-01). For the TTI survey analysis from academic years 2019–2020 and 2021–2022, 48 students completed the CM course, and 297 medical students did not take the CM course.

### 3.1. Competence and Confidence in Diet and Lifestyle Counseling 

There were significant improvements to self-reported confidence/competence in diet and lifestyle counseling variables from pre-course to post-course surveys (*p* < 0.001). Initially, students did not rate themselves as feeling highly confident/competent in diet counseling (pre-course mean score 2.68 +/− 0.63). Immediately following the course, respondents rated themselves as having greater confidence and competence in diet counseling (post-course mean score 4.15 +/− 0.40 *p* < 0.001). Similarly, self-reported confidence/competence in lifestyle medicine counseling increased from pre-course (mean 3.45 +/− 0.49) to post-course (mean = 4.02 +/− 0.57, *p* < 0.001). The results from pre-course and post-course are summarized in Table 5.

### 3.2. Interprofessional Communication 

The post-course survey reflected significant improvements in perceptions of interprofessional communication. Students reported that they were more able to communicate effectively with other professionals in an interdisciplinary manner (MD = 1.20, *p* < 0.001). 

### 3.3. Health Behaviors and Advocacy

Students reported significant improvements in their ability to prepare eight healthy meals (MD = 0.99, *p* < 0.001). Students also perceived themselves as better able to identify their own areas for future growth in the area of nutrition and food preparation (MD = 0.90, *p* < 0.001). Furthermore, the students reported an increased ability to champion a healthy lifestyle and well-being for a community (MD = 0.83, *p* < 0.001). There was no significant change to other health behaviors, including cups of vegetables eaten per day; meals eaten out in the past month; number of hours slept per night; or days per week achieving 30 min of physical activity. Average sleep hours remained around seven hours per night, and days obtaining 30 min of physical activity remained at four days per week. Table 6.

### 3.4. Educational Environment

No significant differences were observed between the in-person or the mixed-mode cooking groups (*p* > 0.05).

### 3.5. What Students Like Best about Course

The post-course survey contained an open-ended question asking students what they liked best about the course. All 84 students commented, and the responses were reviewed for conceptual themes within content, process, and product categories. The text from all comments was also utilized to generate a word frequency cloud with Linguistic Inquiry and Word Count Software (LIWC-22) Figure 1.

#### 3.5.1. Content

Fifty-four students commented on content themes (materials, structure, knowledge, skills, techniques, and organization). Student comments were saturated with an appreciation of the recipes, with 33 out of 54 content comments indicating that they liked how the recipes were “diverse”, “healthy”, “new”, “simple”, “varied”, “high-quality”, and “…taste good while still being healthy”.

#### 3.5.2. Process

Considering the process category, 62 out of 84 students commented on the course’s experiential, social, or relational aspects. Specifically, the hands-on cooking component (26 out of 62), cooking together (19 out of 62), and the interdisciplinary nature of the course (13 out of 62). Several student comments further characterized why they liked the hands-on cooking component: “I appreciated the hands-on cooking experience. I think it’s useless to talk about ways to eat healthier if we do not get to practice it ourselves and experience some of the practical barriers. I felt like that did more to prepare me to empathize with patients and to answer their questions”. Of the many students who emphasized the social aspect of cooking, several described why: “I loved cooking in person with everyone! It really inspired me to cook with friends and family more often”. “I loved getting to cook with people. It showed me that cooking can be fun and, like exercise, it’s worth making time for in your day”. “I…really enjoyed working with people from other fields and cooking, the best team-building activity of all”.

#### 3.5.3. Product

Considering the product category, 40 out of 84 students reported some kind of outcome, change, or application following the course experience. The product comments emerged as two themes that were not mutually exclusive: 25 comments indicated a personal change, and 19 comments mentioned a theme of future clinical application. Those who stated a personal change referred to how: “cooking can be fun, and like exercise, it’s worth making time for in your day” and that “the course gave me a reason and reminder to prepare meals at home”. Others stated how the course shifted their outlook: “It was perfect to read the cases before class, cook, then discuss them over food. It opened my eyes to how easy it can be to discuss nutrition with patients. It does not need to be as intimidating as I once thought”. Mentions of clinical utility emerged as they described tools for future encounters: “I feel more equipped with evidence-based recommendations for my patients regarding nutrition”, and “Motivational interviewing will help me most in the long run and is definitely going to be useful considering we do not learn it in medical school curriculum”. Many comments included personal and clinical notions, such as: “It has helped me to explore my own kitchen and practice concepts of healthy eating in a safe space. I feel so much more confident about my own healthy eating and providing sound dietary advice to others. I also understand the various barriers that get in the way of healthy eating far better and feel well-equipped to discuss behavioral change”. And lastly, “I loved that as I learned about nutrition and cooking, I improved both my personal nutritional habits and my ability to help patients make healthy changes. Practicing the things I will someday recommend to patients makes me feel more able to help them through the challenges of maintaining a healthy diet”.

### 3.6. What Students Liked Least about the Course

The responses to the open-ended survey question regarding what they liked least/what they would change about the course exhibited high variance and did not yield prevalent themes.

### 3.7. TTI Survey Results

Students who took CM were significantly more likely to agree that they could counsel patients about nutrition (*p* < 0.05, r = 0.16) and counsel patients about physical activity (*p* < 0.05, r = 0.13). Students who took the elective during their pre-clinical years were more likely to agree that they could counsel patients about nutrition (*p* < 0.05, r = 0.10) and champion a healthy lifestyle for themselves (*p* < 0.01, r = 0.01). Table 7.

## 4. Discussion

Our findings appeared consistent with other higher education studies evaluating CM courses’ efficacy. Medical students who completed CM courses expressed improved confidence in counseling patients on a healthy diet and higher competency in counseling for lifestyle medicine topics [34,35,38].

The results also highlighted the value that the interprofessional environment added to the U of U CM training. Many CM studies describe the interdisciplinary nature of the instructor teams (MDs, RDNs, and Chefs), but the subjects were mainly medical students. The U of U CM course is offered as an IPE opportunity, making it unique among CM courses. The course features interdisciplinary instructors, an interprofessional peer environment, and includes dietetic student teaching assistants. Moreover, the course uses the framework developed by the Centre for Advancing Collaborative Healthcare & Education (CACHE, formerly CIPE). It has undergone the Process for Interprofessional Education System (PIPES), in which the CM course has become an accredited IPE elective using the University of Toronto’s IPE curriculum standards. The CACHE framework establishes core competencies and values of IPE that contribute to team skills such as communication, conflict resolution, facilitation, accountability, cooperation, interprofessional decision-making, ethics of interdependence, etc. [51]. Several health organizations, including the Institute of Medicine and the World Health Organization, are calling for the wider-scale implementation of Interprofessional Education (IPE) across clinical and educational settings as part of a critical redesign of healthcare systems [52,53]. The increasing complexity of modern healthcare, the frequency of comorbid chronic conditions, and aging populations contribute to over-burdened healthcare providers. PCPs cannot provide high-quality short-term, long-term, and preventive care during a standard 15 min visit nor perform care-coordination functions for which they are not reimbursed. This strain is driving the movement for healthcare to be delivered by teams composed of coordinated professional groups. Coordinated team care may help prevent conflicting care plans, duplication of diagnostic testing, perilous polypharmacy, and reduce physician burnout [54]. In the case of nutrition care from PCPs, a coordinated interprofessional practice model may allow PCPs to provide initial recommendations and oversee treatment plans—avoiding the commonly cited challenge of lack of time to provide specialist knowledge and motivational interventions that elicit behavioral changes [37].

Two questions on the survey pertained to the course’s interdisciplinary nature: (1) how valuable is the inter-disciplinary nature of this class, and (2) students will be able to effectively communicate with other professionals in an interdisciplinary manner. Students rated the first question favorably pre- and post-course, indicating prior positive appraisal of interdisciplinary opportunities. Student responses related to communicating in a professional manner showed a significant improvement from pre-course to post-course. These findings were supported by numerous students’ comments indicating that what they liked best about the course was working with other disciplines.

While a primary goal of a CM course is to build student confidence in counseling patients for successful dietary change, an interrelated goal is for students to develop skills for personal well-being by teaching them how to cook and eat healthfully. Cups of vegetables eaten per day (approximately 2.5 C/d) remained the same pre- to post- course, likely due to high pre-course consumption, yet the students indicated improvements in their ability to prepare eight healthy meals. Likewise, over a third of the U of U CM students described a contemplation of self-change related to their ability to cook more healthfully. These results related to those demonstrated in other CM studies. Hands-on cooking and nutrition education is more successful in improving the quality of student diets than traditional clinical nutrition education [31,37,38,55,56]. Furthermore, prior research indicates that medical students’ and physicians’ personal health habits are significant predictors of their patient counseling practices [39,41,42,44,57]. Medical students with healthier personal habits reported more frequently counseling their patients about prevention [41]. This highlights the importance of provider engagement with their own health behavior as a key to enhancing their patient’s engagement with healthier behaviors. 

Following the course, students rated themselves as more able to identify their own areas for future growth in the area of nutrition and food preparation. They also rated themselves as more able to champion a healthy lifestyle and well-being for a community. For the variable “champion a healthy lifestyle and well-being for myself”, the mean remained the same, somewhere between uncertain and agree.

The secondary purpose of this study was to determine if the educational environment of the hands-on cooking experience significantly impacted the survey responses. No significant differences were found between the groups for the composite or other outcome variables. These results suggested that students still experienced positive outcomes whether they were cooking in person in the teaching kitchen lab or utilizing a combination of cooking at home via Zoom and cooking in-person (mixed-mode). Our findings were promising and should inform future studies that would be able to compare/contrast larger online groups. An example would be a study design that compares four equal groups by educational environment: fully virtual (eight sessions), fully in-person (eight sessions), hybrid (four in-person sessions, four virtual sessions), and a control group that do not participate in a CM course. 

A perceived barrier to implementing a CM elective is the unavailability of a teaching kitchen [58]. For many institutions, a workaround has been collaborating with a community or culinary school kitchen in the area. Our research and others could help CM program coordinators determine whether online CM training is a strategy for overcoming the lack of kitchen infrastructure. Moreover, future studies could investigate the areas of informal feedback that we received from instructors. They described how the virtual sessions eliminated logistical requirements for the teaching staff, who typically purchased ingredients and set up supplies for the hands-on portion. As a tradeoff, students gained comfort in shopping for and preparing recipes in their own cooking spaces. 

The qualitative analysis was helpful to gain insight into how students experienced the course in relation to its design. Students recorded highly positive responses to all aspects of the course content, course process, and many students synthesized a course “product.” They appreciated preparing and learning about healthy, simple, diverse recipes; conversations with other students in other disciplines; cooking with others; and cooking hands-on in the kitchen. Of note was the perceived positive influence on their future clinical practice and self-care concerning healthy eating. Students synthesized greater self-motivation in these areas, adding insight to this study’s significant quantitative results. Studies that have previously assessed the nature and quality of culinary nutrition education found that the social and experiential aspects drive positive changes in eating behaviors [59]. They note that program experiences, which included the common themes of collaboration, celebration, skill building, skill reinforcement, and peer support, among other experiential drivers, were the most effective in motivating behavioral changes [60]. 

The data from the TTI survey allow a comparison to be made between medical students who participated in the CM elective and those who did not participate during their tenure in medical school. Medical students who completed the CM course were more likely to strongly agree that they could counsel patients on nutrition and physical activity. Of note is that medical students who took the CM course in their pre-clinical years (years 1 or 2) rated their ability to counsel on physical activity higher than those who took CM in their clinical years (years 3 or 4). This was also the case for students’ ability to champion healthy lifestyle and well-being for themselves. While the effect size was small, these results showed a novel trend for students’ abilities to counsel patients and their abilities to champion healthy lifestyles for themselves throughout medical school. This longitudinal data have yet to be observed in the literature and so requires further investigation. We hypothesize that integration of a CM curriculum early in medical training has a greater potential for positively affecting well-being of medical students for the duration of their medical training. 

This study has limitations shared by other CM studies. As CM is an emerging field, the courses are often electives at single institutions. As such, our limitations are small sample size (*n* = 84), data collected from only a single site, and participant self-selection bias due to the elective nature of the course—and, presumably, student pre-interest in nutrition and lifestyle medicine. Additionally, readers should exercise caution that our pre/post-course survey responses were treated as between-subjects data due to the anonymity of surveys. Thus, the mean differences between pre- and post-course surveys may not reflect within-person changes. However, after controlling for a group interaction and considering the size of the effects within the analysis, there is potential that the course had an impact. 

A strength of this study was the qualitative analysis of what students liked best about the course. Student comments reflected the physical, intellectual, social, and emotional experiences that led them to improved self-perceptions. Their feedback provided insight into what influenced their learning and inclined them to report quantitative changes in their confidence and competence. 

For future assessment of specific variables such as health behaviors, attitudes, strength of IPE, or influence on future practice, enhancements to the pre/post-course survey are warranted. CM research could also be supported by the development of validated surveys and tools that objectively measure counseling proficiency and universally preferred CM outcomes. Lastly, future CM research is needed to examine if and how positive outcomes persist and how they translate into strengthening clinical practice through a nutritional and lifestyle medicine approach. 

## 5. Conclusions

This study indicates that the U of U CM course improved perceived dietary and lifestyle counseling skills for medical and other health professional students. Students also improved their self-reported ability to communicate with other disciplines in a professional manner, identify their own areas for future growth in nutrition and food preparation, and advocate for healthy lifestyles and wellness for a community. Furthermore, students rated themselves as more able to prepare eight healthy meals. The course’s online and semi-online (mixed-mode) cooking experiences appeared to convey the same benefits to students as the in-person cooking experience in a kitchen lab. A key facilitator for course likeability was the recipe collection. Other key facilitators were the experiential or relational opportunities for hands-on cooking, cooking with others, and interacting with other disciplines. These experiences channeled into an appraisal of greater comfort and awareness of how to improve their eating habits and help their patients make healthy changes. These results suggests that the CM course provided the imperative contextual learning experiences that resulted in students’ greater sense of self-efficacy for counseling patients about diet and lifestyle, advocating for wellness, and preparing healthy recipes. Likewise, participation in the CM course may potentially strengthen their clinical care with nutritional and team-based approaches.

## Figures and Tables

**Figure 1 nutrients-15-04157-f001:**
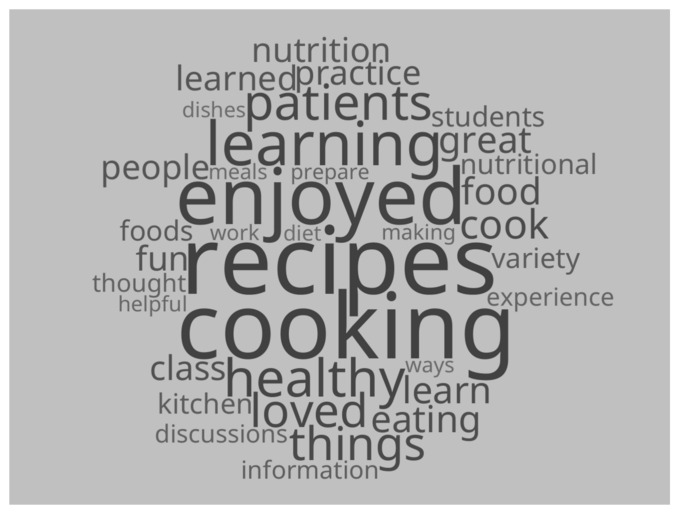
LIWC-22 Word frequency cloud generated from students’ comments about what they liked best about the U of U CM course. © Britta Retzlaff Brennan.

**Table 1 nutrients-15-04157-t001:** Educational objectives for the U of U Culinary Medicine course.

EO1	Describe the components of a Mediterranean diet
EO2	Identify strengths and weaknesses in a patient’s diet
EO3	Convey concise dietary advice to simulated patients
EO4	Prepare at least eight healthy meals
EO5	Understand the role that diverse disciplines play on a healthcare team
EO6	Identify priorities for growth in the area of nutrition and basic culinary skills
EO7	Demonstrate ability to obtain basic nutrition history and use basic motivational interviewing skills

**Table 2 nutrients-15-04157-t002:** Course outline and curriculum for 8-week U of U Culinary Medicine elective.

Week	Topic	Activities	Assessment
1	Introduction to Culinary and Knife Skills	Pre-course survey/self-assessment (EO6) Video lectures: Intro to CM, Nutrition 101, Dietary Patterns Article: Mediterranean diet and health status: Active ingredients and pharmacological mechanisms Case Study: High LDL (EO1) COOKING LAB: Mediterranean (EO1, EO4)	Quiz 1
2	Behavior Change and Diet History	Video lectures: Behavior Change, Motivational Interviews Article: Collaboration and Negotiation: The Key to Therapeutic Lifestyle Change Case Study: Hypertension (EO1, EO2) COOKING LAB: Southwestern (EO3, EO4)	Quiz 2
3	Weight Management and Eating Disorders	Video Lectures: Weight Management, Strategies for Weight Change, Eating Disorders Articles: US Preventative Task Force Weight Loss Interventions 2018, American Psychiatric Diagnostic Criteria Case Study: Weight Management (EO7) COOKING LAB: Eastern Mediterranean (EO4)	Quiz 3
4	Diabetes and Carbohydrates	Video Lectures: Diabetes, Sleep, Physical Activity Article: Exercise is Medicine: From a Vital Sign to Vitality: Selling Exercise So Patients Want to Buy It Case Study: Elevated A1c, hypercholesteremia, hypertension (EO2) COOKING LAB: Whole Grain Breakfasts (EO4)	Quiz 4
5	Food Sensitivity, Elimination Diet, and Irritable Bowel Syndrome	Video Lectures: IBS, Food Sensitivities, Microbiome Article: Food: The Main Course to Wellness and Illness in Patients with Irritable Bowel Syndrome Case Study: Celiac, FODMAPS, Anti-Inflammatory Diet COOKING LAB: Indian (EO4)	Quiz 5
6	Renal Disease and Cardiovascular Disease	Video Lecture: CKD, Label Reading, and Eating on a Budget Case Study: Local resources for food insecurity COOKING LAB: Asian Curries and Salads (EO4)	Quiz 6
7	Pediatrics and Adolescents	Video Lectures: Kids and Food, Kids and Lifestyle Article: How to Teach Children about Healthy Eating without Food Shaming Case Study: Healthy dinners for families COOKING LAB: Optimizing traditional entrees (EO4)	Quiz 7
8	Micronutrient Insufficiencies, Supplements, and Sports Nutrition	Video Lectures: Sports Nutrition, Supplementation Article: Physical Activity Counseling in Primary Care: Insights from Public Health and Behavioral Economics Case Study: Athlete Nutrition (EO3) COOKING LAB: Create your own pantry meal (EO4)	Quiz 8

**Table 3 nutrients-15-04157-t003:** Group descriptions by semester and cooking environment.

Groups	In-Person	Hybrid ^1,^*	Online ^2,^*
Semesters	Fall 2019 Spring 2020 Fall 2022 Spring 2023	Fall 2020 Fall 2021 Spring 2022	Spring 2021
Students (*n*)	*n* = 47	*n* = 31	*n* = 6

^1^ In-person cooking and at-home cooking via Zoom. ^2^ At-home cooking via Zoom. * Hybrid and online groups combined into “mixed-mode” group for analysis.

**Table 4 nutrients-15-04157-t004:** Survey questions included in the composite variables for confidence/competence in dietary counseling and competence in lifestyle medicine counseling.

Composite Variables	Survey Questions
Confidence/Competence in Dietary Counseling	Feel confident I could advise patients about dietary change Able to identify strengths and weaknesses in patient’s diet
	Able to convey concise dietary advice to patients Able to describe the components of a Mediterranean diet Able to help patients make behavior change
Competence in Lifestyle Medicine Counseling	Able to help patients utilize full breadth of possibilities to prevent illness before it starts Able to counsel about physical activity Able to counsel about sleep Able to counsel patients about stress Able to counsel patients about natural products

**Table 5 nutrients-15-04157-t005:** Pre-course to post-course changes in students’ perceived confidence/competence in dietary counseling and lifestyle medicine counseling.

	Cronbach’s Alpha	Mean		*p*-Value
		Pre-	Post-	
Self-reported confidence and competence in dietary counseling	0.65	2.68	4.18	<0.001
Self-reported competence in lifestyle medicine counseling	0.66	3.45	4.12	<0.001

**Table 6 nutrients-15-04157-t006:** Students’ pre-course to post-course changes.

Survey Questions		Mean		
	SD	Pre- **	Post- **	*p*-Value
Interprofessional Environment				
How valuable is the inter-disciplinary nature of this class (students potentially from colleges of health, nursing, pharmacy, and school of medicine):	+/−0.77	4.59	4.40	0.07
Students will be able to effectively communicate with other professionals in an interdisciplinary manner.	+/−0.89	3.72	4.48	<0.001
Health Behaviors and Advocacy				
Students will be able to identify their own areas for future growth in the area of nutrition and food preparation.	+/−0.87	3.59	4.49	<0.001
Champion a healthy lifestyle and well-being for myself.	+/−0.81	3.80	4.15	0.005
Champion a healthy lifestyle and well-being for a community.	+/−0.82	3.46	4.29	<0.001
Students will be able to prepare eight healthy meals.	+/−1.09	3.58	4.57	<0.001
How many cups of vegetables do you eat, on average, each day? *	+/−1.20	2.24	2.69	0.09
How many days of the week, on average, did you get at least 30 min of moderate physical activity? *	+/−1.91	3.80	4.13	0.29
How much sleep do you get on an average night? *	+/−1.08	6.92	7.10	0.35
How many meals did you eat out (i.e., restaurant, cafeteria, store, vending machine, etc.)? *	+/−3.17	6.37	5.55	0.10

* Not Likert scale variables; ** Mean +/− SD.

**Table 7 nutrients-15-04157-t007:** Comparing medical students who took the U of U CM course to medical students who did not participate in the CM course in counseling for nutrition and physical activity and comparing CM students by year in which they took the CM course (pre-clinical or clinical) in nutrition counseling and well-being self-advocacy.

TTI Survey Question	Participation in U of U CM Course	Mean	*p*-Value
I agree I can counsel patients about nutrition	No (*n* = 297)	3.67	*p* = 0.025
	Yes (*n* = 48)	4.17	
I agree I can counsel patients about physical activity	No (*n* = 297)	4.17	*p* = 0.016
	Yes (*n* = 48)	4.46	
I agree I can counsel patients about physical activity	Year 1 or 2 (pre-clinical)	4.51	*p* = 0.036
	Year 3 or 4 (Clinical)	4.15	
I agree I can currently champion a healthy lifestyle and well-being for myself (scale was 1–10)	Year 1 or 2 (pre-clinical)	6	*p* = 0.009
	Year 3 or 4 (Clinical)	4.4	

## Data Availability

Data available on request due to restrictions (e.g., privacy or ethical).

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
