# Peer review of "Impact of Culinary Medicine Course on Confidence and Competence in Diet and Lifestyle Counseling, Interprofessional Communication, and Health Behaviors and Advocacy"

_nutrients, 2023, doi:10.3390/nu15194157_

Round 1

Reviewer 1 Report

This is a very interesting article examining an important area, especially with regards to nutrition education programs in clinical education programs.  It is well-written.  

This reviewer has two primary comments

1.  There should be a purpose statement at the end of the introduction section.  

2.  Lines 178-192 in the methods section should be incorporated into the discussion.  

Additionally, please comment on the bias of the sample based on who selected to take the CM course and who didn't.  

Reviewer 2 Report

The potency of diet and its influence on health calls for actions to improve confidence and competence in diet and lifestyle counseling by health professionals. The proposition of an interprofessional culinary medicine course for medical and health professional students is an innovative and exciting strategy to enhance the overall diet quality of individuals treated by healthcare professionals responsible for these issues. Considering that, it is recommended to promote investigation on this topic. Despite this, it is tough to measure and describe how an innovative proposition may impact professional practices, including perceived knowledge, skills, attitudes, and other attributes required to enact professional practice competently within a given scenario to build student confidence in counseling patients for successful dietary change. The complex and multiple dimensions of food call for initiatives that discuss this aspect during change;

- Method: Needs to be revised (the analyses for the pre-/post-course need to consider separately each group's descriptions by cooking environment due to its differences. Furthermore, It is not clear the theoretical framework used to select the chosen analyzed variables/constructs, as shown in the results section: Competence and Confidence in Diet and Lifestyle Counseling, Perceptions of Interprofessional communication and well-being advocacy, wellness behaviors, group and time interactions, what students like best about course).

Discussion and other manuscript sections must be revised following previous

comments.
